# Highland Barley Replaces Sorghum as Raw Material to Make Shanxi Aged Vinegar

**Huan Zhang [1,2], Jingli Zhou [3], Fanfan Lang [3], Yu Zheng [4] and Fusheng Chen [1,2,*]**

[1] Hubei International Scientific and Technological Cooperation Base of Traditional Fermented Foods, Huazhong Agricultural University, Wuhan 430070, China; hz863@hotmail.com

[2] College of Food Science and Technology, Huazhong Agricultural University, Wuhan 430070, China

[3] Shanxi Zilin Vinegar Industry Co., Ltd., Taiyuan 030400, China; zhoujingli1110@126.com (J.Z.); jszxlff@zlcy.com (F.L.)

[4] State Key Laboratory of Food Nutrition and Safety, College of Biotechnology, Tianjin University of Science & Technology, Tianjin 300457, China; yuzheng@tust.edu.cn

* Correspondence: chenfs@mail.hzau.edu.cn

**Abstract:** Highland barley (HB, *Hordeum vulgare* L. var. *nudum* Hook. f.), also known as naked or hulless barley, is a kind of cereal crop growing at high altitudes (4200–4500 m) around the world. In this study, HB vinegar (HBV) was prepared, using Tibetan HB as the main raw material, according to the process of Shanxi aged vinegar (SAV), a famous vinegar in China, in which sorghum is usually used as the main raw material. The related main compounds, such as alcohol and acetic acid, in the alcohol and acetic acid fermentation processes were monitored and analyzed, respectively. The flavor components in the aged vinegars were analyzed by headspace solid-phase microextraction, combined with gas chromatography-mass spectrometry, and compared with sorghum vinegar (SV), which was made, using sorghum as the raw material, based on the SAV process. The results revealed that at the alcohol fermentation stage, the alcohol content of HB mash was higher than that of the sorghum mash ($p < 0.05$), and at the acetic acid fermentation stages of HBV and SV, the total acid contents were 6.23 and 5.81 (g·100 mL$^{-1}$ and $p < 0.05$), respectively. After aging one and a half years, the contents of non-volatile acid, volatile acid, and ester compounds in HBV were higher than those in SV. Therefore, HB can replace sorghum as the raw material for making SAV. Based on a literature search, the comparison and analysis of the main components and volatile flavor compounds of HBV and SV were not studied before.

**Keywords:** highland barley; Shanxi aged vinegar; flavor compounds; gas chromatography-mass spectrometry

## 1. Introduction

Highland barley (HB, *Hordeum vulgare* L. var. *nudum* Hook. f.), also called naked or hulless barley (Qingke, in Chinese), is one of the variations of the Gramineae wheat family [1]. As a type of cereal crop grown in high altitudes (4200–4500 m) around the world, HB is characterized by early maturity, cold resistance, stable yield and wide adaptability [1,2]. In recent years, the planting area and yield of HB in China have increased steadily, reaching 25000 hectares and 1.1 million tons in 2019, respectively [3]. Besides starch, HB is also rich in β-glucan, dietary fiber, phenolic compounds, vitamin E, and other nutrients that are beneficial to human health [4–6]. Research has showed that HB possesses the functions of anti-oxidation, lowering blood pressure, adjusting blood lipid, and modifying intestinal flora, etc. [7–10]. HB has been produced into various foods such as noodles, HB wine, beverages, and so on [11–13]. Thus far, there has only been a few reports on HB used to brew vinegar [14]. In addition, based on our literature search, the comparison and analysis of main components and volatile flavor compounds (VFCs) of aged HBV and SV has not been done before.

Vinegar, as a sour seasoning, is widely used in diets around the world. Due to the differences in regions, raw materials, and technologies, there are distinct varieties of vinegar in different countries and regions [15]. Generally, in Western countries such as Italy, Spain, Germany, the United States, and so on, vinegars are made mainly from fruits, including grape and apple, through liquid-state fermentation, and, thus, are called fruit vinegars [16]. In Eastern countries, such as China, Japan, and Korea, etc., vinegars are made mainly from sorghum, rice, corn, wheat or other cereals rich in starch by solid-state fermentation (SFP), and, thus, are called cereal vinegars [17]. In China, there are various kinds of cereal vinegars [18]. Among them, Shanxi aged vinegar (SAV) is one of the most famous cereal vinegars produced by SFP, and it is usually produced with sorghum as the main raw material, and bran, rice or millet husk as auxiliary materials. In the SAV process, some unique processes occur, for instance, after acetic acid fermentation (AAF), about 30% vinegar paste (called *Cupei*, 醋醅 in Chinese) is heated, and new (fresh) vinegar is aged at least half a year before bottling and entering the market [19]. All these unique processes give SAV its characteristic thickness, mellow sourness, and delicate fragrance, making it different from other vinegars [20,21].

In the current study, HB replaced sorghum to make HB vinegar (HBV) based on the SAV process. The composition of HB and sorghum was analyzed, changes of main compounds including alcohol and total acid, etc. during the production processes of HBV and sorghum vinegar (SV) were detected and compared, and the VFCs of HBV and SV, aged one and a half years, were determined by headspace solid phase microextraction-gas chromatography-mass spectrometry (HS-SPME/GC-MS). The results showed that HB is an alternative raw material that can be used to make the cereal vinegar.

## 2. Materials and Methods

### 2.1. Raw Materials

Tibetan highland barley (HB), sorghum, and starters, including *Daqu*, *Kuaiqu* and *Jiumu*, were provided by Shanxi Zilin Vinegar Industry Co., Ltd., Shanxi Province, China. *Daqu*, which contains various hydrolases such as amylase and protease, is made of a mixture of barley, wheat, and pea by spontaneous fermentation [22]. *Kuaiqu*, which contains high amylase and glucoamylase activity, is a type of starter prepared with *Aspergillus niger* [23]. *Jiumu* is another starter containing a large number of yeasts, such as *Saccharomyces cerevisiae* [24].

### 2.2. The Process of Vinegar Production

The production process (Figure 1) of vinegars in the current study includes saccharification, alcoholic fermentation (AF), acetic acid fermentation (AAF), heating, drenching, and aging based on the SAV process [24].

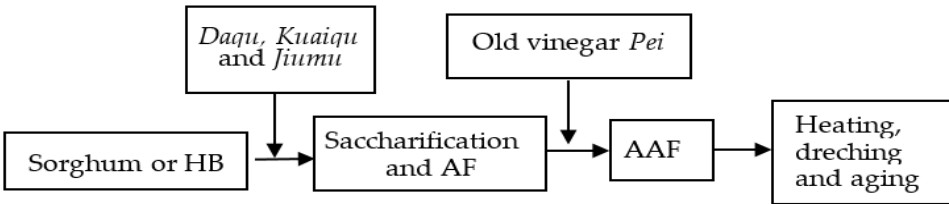

**Figure 1.** The process flow chart for Shangxi aged vinegar.

Briefly, after the raw material (HB or sorghum) is crushed into pieces, it is placed in a ceramic container, and then mixed evenly with *Daqu* and *Kuaiqu* powder, *Jiumu*, and water to make the fermentative mash according to the ingredient ratios listed in Table 1. During the first three days, the mash is stirred and mixed twice a day, and further standing fermentation occurs until 16 days (AF). After that, the vinegar paste is made by mingling the mash evenly with 100 kg of raw materials, including 50 kg of bran (Table 1) and the old vinegar paste (including acetic acid bacteria) from the last production, to start AAF for

10 days in the ceramic container, turning over the vinegar paste twice a day. Then, 30% vinegar paste is heated for 5 days, during which the temperature of the vinegar paste is gradually increased from 30 °C to 90 °C in the first 3 days, then decreased to 30 °C on the fifth day, resulting in a color change from light yellow to brown. Finally, the heated vinegar paste is blended with the unheated vinegar paste (about 70%), and leached with saline to obtain the new (fresh) vinegar, which is aged for one and a half years in an open pottery container, into aged vinegar.

**Table 1.** Ingredient proportion for vinegar production.

| Categories | Ingredient Proportion (Kg) | | | | |
|---|---|---|---|---|---|
| | Raw Material | *Daqu* | *Kuaiqu* | *Jiumu* | Material: $H_2O$ | Bran |
| **HBV** | 100 (HB) | 45 | 15 | 0.20 | 1:4.0 | 50/100 HB |
| **SV** | 100 (Sorghum) | 45 | 15 | 0.20 | 1:4.0 | 50/100 Sorghum |

HBV: Highland barley vinegar; and SV: Sorghum vinegar.

### 2.3. Determination of Physicochemical Parameters

2.3.1. β-glucan Determination

The β-glucan contents of the samples were analyzed by the Megazyme mixed β-glucan detection kit produced by Beijing Micro Wise Co., Ltd., Beijing, China, following the manufacturer's instructions. In brief, the process included sample preparation and enzymatic hydrolysis and β-glucan determination.

Sample preparation: for HB and sorghum, a 0.2 g (100 mesh) sample was treated with a 10 mL 50% ethanol solution for 10 min at 80 °C in the water bath, and then centrifuged at 2000 rpm for 10 min. The centrifugal sediment was treated once with 50% ethanol solution at the above-mentioned conditions. The final precipitate was used for the next enzymatic hydrolysis and β-glucan determination. For the alcoholic fermentation mashes and vinegars, three times the volume (15 mL) of 95% ($V/V$) ethanol solution was added into a 5 mL sample. After shaking violently, the mixture stood for 5 min, and was centrifuged at 2000 rpm for 10 min. The centrifugal sediment was treated with a 10 mL 50% ethanol solution, and the mixture was centrifuged at 2000 rpm for 10 min to get the final precipitate for enzymatic hydrolysis and β-glucan determination.

Enzymatic hydrolysis and β-glucan determination: the precipitate from the sample preparation was dispersed into a 0.2 mL 50% ethanol solution by shaking violently. After that, 4 mL of phosphate buffer (20 mM, pH6.5) was added and put into boiling water for 5 min to completely dissolve the precipitate. Then, the mixture was moved to the water bath at 50 °C for about 10 min, and a 0.2 mL 50 U/mL lichenase solution from the kit was supplemented, mixed well, and kept for 60 min in the water bath. A 5 mL acetate buffer (200 mM, pH 4.0) was then supplemented, mixed well, and centrifuged at 2000 rpm for 10 min. Then, 1 mL of supernatant and 0.1 mL 2 U/mL β-glucosidase solution from the kit were mixed evenly, and kept at 50 °C for 10 min. The glucose content in the mixture was detected by glucose oxidase, and the β-glucan contents of samples were calculated by the following formula:

$$H = \Delta A \times F \times 9.4 \times 10^{-6}/W \times 0.9 \times 100$$

where H is the percentage of β-glucan in the sample; $\Delta A$ is the difference between the absorbance values of the sample and the blank; F is the ratio of 100 μg glucose divided by the absorbance value of 100μg glucose; 9.4 is the volume correction factor; W is the sample weight or volume (g or mL); and 0.9 is the coefficient of glucose condensation into β-glucan.

2.3.2. Total Flavonoids Analysis

Total flavonoids content (TFC) of the vinegar samples was detected by using a colorimetric assay, with minor modifications, according to the method described in the literature [25].

Briefly, 2.5 mL of vinegar and 0.3 mL of $NaNO_2$ (5%) were mixed. After 6 min, 0.3 mL of $Al(NO_3)_3$ (10%) was added into the mixture, mixed well, and left to stand for 6 min. Next, 4 mL of NaOH (4%) was added into the mixture, then the mixture volume was set to 25 mL with distilled water, and blended well. The absorbance was measured at 510 nm. Rutin was used as the standard and the tests were performed in triplicate.

### 2.3.3. Total Acid and Ester Analyses

The total acid (TA) and total ester (TE) were analyzed following the continuous potentiometric titration method described in the literature [26].

For TA analysis, 10 mL of vinegar was put into a 100 mL volumetric flask, and the volume was set to 100 mL with distilled water. Then, 20 mL of solution was taken from the volumetric flask to a 250 mL beaker, and 60 mL of distilled water was added and mixed. Then, a 0.1 M NaOH solution was used to titrate the pH value to 8.2, which was monitored by the pH meter (PHS-25, Shanghai Yidian Scientific Instrument Co., Ltd., Shanghai, China). Finally, the TA contents of the samples were calculated based on 0.1 M NaOH standard solution consumption.

For TE determination, we transferred the titrated vinegar sample from the TA analysis into a 350 mL round bottom beaker, added 25 mL of 0.1 M NaOH solution, and mixed well. After a condenser was connected with the beaker, the mixture was reacted (saponified) for 0.5 h in boiling water. After the mixture was cooled to room temperature and moved to a 250 mL beaker, we used a 0.1 M $H_2SO_4$ solution to titrate it to pH 9.5. The TE contents of the samples were calculated based on 0.1 M $H_2SO_4$ solution consumption. The TE contents were expressed as the content of ethyl acetate.

### 2.3.4. Other Physicochemical Indexes Determination

The determination of starch, protein, and fat in raw materials (HB and sorghum), according to the literature [27], and the amount of alcohol, amino acid nitrogen, reducing sugar, etc. in the alcohol fermentation mash and vinegar paste were detected by the methods described in the literature [28–32].

### 2.4. Analyses of Volatile Flavor Compounds in Vinegars

The VFCs of vinegars were extracted by HS-SPME (Supelco, Bellefonte PA, USA) and determined by GC-MS/MS (Shimadzu QP2010, Shimadzu Corporation, Kyoto, Japan). To do so, 2 mL of vinegar and 6.0 mL of distilled water were placed in a 20 mL headspace bottle; 1 g of NaCl and a magnetic agitator were added into the bottle, then the bottle cap was tightly screwed into place. After the solution was agitated and equilibrated at 60 °C for 5 min, the dimethylsiloxane (DVB/CAR/PDMS) fiber was inserted into the bottle through the cap and headspace absorption was performed for 40 min. Upon completion, the fiber was inserted into the injection port (220 °C) of the GC-MS instrument to desorb for 5 min. The carrier gas was pure helium with flow rate of 1 mL/min with a separation ratio of 5:1. The initial temperature of the program, 40 °C, was increased to 150 °C at a rate of 4 °C/min, then increased to 250 °C at a rate of 8 °C/min and kept for 6 min. MS conditions: the EI ionization source temperature was 230 °C, the energy was 70 eV, and the scanning range was set at 35–500 $m/z$. The VFCs were identified by comparing their mass spectra with those from the NIST 11 library, and the concentrations of the VFCs were estimated by the concentration of the internal standard substance (2-Octanol) and its peak area with compounds [33].

### 2.5. Statistical Analysis

The experiments were performed in three repetitions and the data were expressed as the mean $\pm$ standard deviation (SD). The significance of the differences between groups was analyzed by the analysis of variance (ANOVA) using GraphPad Prism version 8.0 (GraphPad Software, San Diego, CA, USA), and a $p$-value of <0.05 was considered as the threshold for statistical significance. The heatmap was drawn by TBtools [34].

## 3. Results and Discussion

### 3.1. Ingredients Analyses of Raw Materials

The main components in HB and sorghum, including starch, protein, fat and β-glucan, were analyzed. The results (Figure 2) revealed that the contents of starch in HB and sorghum were high, up to 75.47% and 68.06%, respectively, while the contents of protein and fat in both raw materials were relatively low by comparison. Moreover, HB also contained 5.72% β-glucan, but the sorghum did not contain β-glucan. Compared with sorghum, HB contained the higher content of starch and protein, and, thus, HB may be more suitable as the raw material of SAV.

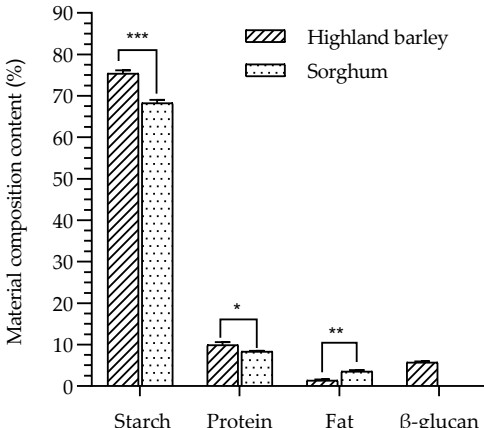

**Figure 2.** Contents of main substances in highland barley and sorghum; * $p < 0.05$; ** $p < 0.01$; and *** $p < 0.001$.

### 3.2. Constituent Analyses during Vinegar Production

#### 3.2.1. Constituents in Alcohol Fermentation Mashes

Alcohol fermentation (AF) is one of the processes in SAV production. During AF, the starch is converted into a reducing sugar (RS) by amylase and glucoamylase from *Daqu* and *Kuaiqu*, then the RS is converted into alcohol, which is the main substrate for AAF, by *Saccharomyces cerevisiae* from *Jiumu* [35]. The component contents in the mashes of HB and sorghum at end of AF (16 days) are shown in Table 2.

**Table 2.** The component contents in alcohol fermenting mashes of HB and sorghum.

| Categories | Alcohol (%) | TA (g·100 mL$^{-1}$) | Residual Starch (g·100 mL$^{-1}$) | Residual RS (g·100 mL$^{-1}$) | β-glucan (mg·L$^{-1}$) |
|---|---|---|---|---|---|
| HB AF mash | 8.35 ± 0.07 | 2.36 ± 0.04 | 2.49 ± 0.01 | 0.44 ± 0.01 | 6.74 ± 0.08 |
| Sorghum AF mash | 7.67 ± 0.18 | 2.32 ± 0.07 | 2.02 ± 0.02 | 0.30 ± 0.02 | - |

HB: Highland barley; AF: alcohol fermentation; TA: total acid; RS: reducing sugar; -: none detected. Values are presented as means ± standard deviations (*n* = 3).

Compared with the sorghum AF mash, the contents of alcohol and residual starch were higher in HB AF mash ($p < 0.05$), while the contents of TA and residual RS were just a little higher in the HB AF mash ($p > 0.05$; Table 2). A trace amount of β-glucan (6.74 mg·L$^{-1}$) was also detected in the HB AF mash, but none was detected in the sorghum AF mash (Table 2).

#### 3.2.2. Changes of Alcoholicity and TA Contents during AAF

Acetic acid fermentation (AAF) is another key process for vinegar production, which is the main stage for acetic acid production and also for flavor formation [23,36]. The changes of alcohol and TA concentrations during AAF of HBV and SV were analyzed and are shown in Figure 3.

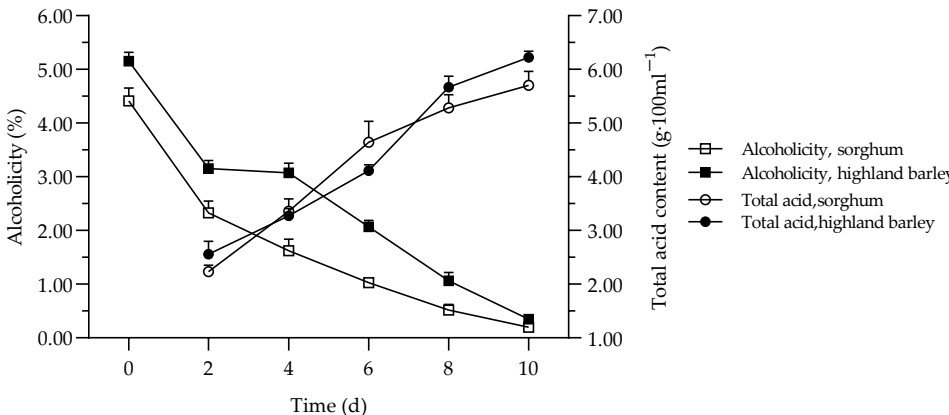

**Figure 3.** Changes of alcohol and TA concentrations during AAF.

As expected, during the AAF processes, the alcohol and TA concentrations showed downward and upward trends, respectively (Figure 3). At the end of AAF (at 10th day), the TA contents reached up 6.23 and 5.81 (g·100 mL$^{-1}$) in HBV and SV pastes, respectively, there was a significant difference ($p < 0.05$), and the alcohols were almost exhausted in both vinegar pastes, with less than 0.5% alcohol left.

### 3.2.3. Analyses of the Main Components in New and Aged Vinegars

Aging in an open pottery container is a unique process of SAV production [19]. In the ageing process of SAV, with the evaporation of volatile substances and water, the volume and viscosity (thickness) of the vinegar decreases and increases, respectively, and the pungent taste reduces, which makes the vinegar more mellow, soft, and harmonious. Meanwhile, due to the esterification and Maillard reactions, various components in vinegar also change considerably [37]. The main composition of the new (fresh) and aged (one and a half years) HBV and SV were determined and compared (Table 3).

**Table 3.** The main composition in HBV and SV before and after aging.

| Items | New Vinegars | | Aged Vinegars | |
|---|---|---|---|---|
| | **HBV** | **SV** | **HBV** | **SV** |
| TA (g·100 mL$^{-1}$) | 4.94 ± 0.04 | 4.93 ± 0.02 | 6.65 ± 0.06 | 6.55 ± 0.05 |
| Non-volatile acid (g·100 mL$^{-1}$) | 1.62 ± 0.01 | 1.66 ± 0.02 | 2.75 ± 0.04 | 1.94 ± 0.02 |
| Amino acid nitrogen (g·100 mL$^{-1}$) | 0.20 ± 0.01 | 0.21 ± 0.02 | 0.31 ± 0.04 | 0.31 ± 0.02 |
| RS (g·100 mL$^{-1}$) | **1.24 ± 0.01** | **1.24 ± 0.01** | **0.99 ± 0.02** | **0.88 ± 0.04** |
| Salt (g·100 mL$^{-1}$) | 0.89 ± 0.02 | 0.82 ± 0.03 | 1.03 ± 0.03 | 1.53 ± 0.03 |
| Soluble salt-free solid (g·100 mL$^{-1}$) | 9.46 ± 0.05 | 8.16 ± 0.02 | 14.10 ± 0.11 | 13.64 ± 0.04 |
| TE (g·100 mL$^{-1}$) | **3.13 ± 0.06** | **3.50 ± 0.03** | **2.70 ± 0.02** | **3.35 ± 0.05** |
| TF (mg·100 mL$^{-1}$) | 73.91 ± 0.19 | 88.71 ± 0.03 | 109.00 ± 1.41 | 113.00 ± 1.39 |
| Ligustrazine (mg·100 mL$^{-1}$) | 3.05 ± 0.05 | 3.35 ± 0.04 | 5.49 ± 0.08 | 9.13 ± 0.02 |
| β-glucan | - | - | - | - |
| Volume (L) | **100.00 ± 0.71** | **100.00 ± 0.98** | **68.00 ± 0.69** | **69.00 ± 0.92** |

HBV: Highland barley vinegar; SV: Sorghum vinegar; New vinegar: unaged vinegar; Aged vinegar: aged for one and a half years; TA: total acid; RS: reducing sugar; TE: total ester; TF: total flavonoid; -: Not detected. Values are presented as means ± standard deviations (*n* = 3). The numbers in bold in the table indicate that the index values in aged vinegars are less than those in new vinegars.

The results (Table 3) revealed that, compared with the new vinegars, with the exception of RS and TE, the main components of the aged vinegars were improved to a certain extent. At the end of ageing, due to the evaporation of volatile components and water in vinegars, the volumes of aged HBV and SV were only about two-thirds of those of the new vinegars, and the viscosity and consistency of the aged vinegars were also improved to a great extent. The RS reduction in aged vinegars may be caused by the RS consumption of the Maillard reaction during the ageing process, while the TE decreases may be due to its

high volatility [38]. Both the TA and non-volatile acid in the aged vinegars were greatly improved. The TA increase may be mainly caused by the increase of the contents of the non-volatile acids [39]. A higher concentration of non-volatile acids, such as lactic acid, succinic acid, and gluconic acid, together with a higher content of soluble salt-free solid can neutralize and reduce the pungent smell of volatile acids, such as acetic acid, so as to make the vinegars more soft and harmonious [40,41]. The characteristic component of SAV, ligustrazine (2,3,5,6-tetramethylpyrazine), also one of the products of the Maillard reaction [42], and TF, both of which have many physiological functions, such as dilating blood vessels, lowering blood pressure, and improving cerebral blood circulation, anti-cancer, anti-bacteria, anti-virus, and antioxidant functions, etc. [43–46], were also greatly improved after aging. Ligustrazine and TF are considered to be the material basis of various physiological functions of SAV [47,48].

β-glucan is one of the main compounds, and the most widely studied bioactive ingredient, in HB, which can affect energy metabolism, lower blood glucose, and increase insulin response [7]. Unfortunately, β-glucan was not detected in the new and old vinegars (Table 3). In particular, in HBV prepared by HB, with a high content of β-glucan (5.72%, Figure 2), β-glucan was not detected even though a trace of β-glucan (6.74 mg·L$^{-1}$) was detected in HB AF mash at the end of the AF stage (Table 2). This may be due to the poor water-solubility of β-glucan, meaning that most of the β-glucan was left in the vinegar residue, while a small amount of water-soluble β-glucan might be digested by microorganisms from *Daqu* and other starters [49].

Compared with the aged SV, the contents of TA, RS, salt, soluble salt-free solid, and TE had no significant difference ($p > 0.05$), while the content of non-volatile acid in the aged HBV was significantly ($p < 0.05$) high and the contents of TE, TF, and ligustrazine were significantly ($p < 0.05$) lower (Table 3).

### 3.3. Volatile Flavor Compounds Analysis of Aged Vinegars

HS-SPME/GC-MS was used to determine the VFCs in the aged HBV and SV. Acetic acid as the main compound and also the very important VFC in vinegars, which the detection results had great inaccuracy by HS-SPME/GC-MS due to the low efficiencies of solid phase extraction and desorption [50]. Therefore, acetic acid was not included in the VFCs of HBV and SV (Figure 4; Supplementary Materials, Table S1).

Based on the VFCs in the aged HBV and SV by HS-SPME/GC-MS (Table S1), a total of 102 VFCs were found in both vinegars, including 60 common VFCs, and 24 and 18 unique VFCs from the aged HBV and SV, respectively, which were mainly classified into acids, alcohols, esters, aldehydes, furans, ketones, and pyrazines, etc. (Table 4; Table S1).

**Table 4.** VFCs classification in aged HBV and SV.

| Species | Concentrations (mg·L$^{-1}$) | |
|---|---|---|
| | **HBV** | **SV** |
| Acids | 2114.14 ± 30.50/(11) | 1644.47 ± 36.39/(12) |
| Alcohols | 2432.88 ± 59.03/(14) | 4415.19 ± 94.08/(12) |
| Esters | 2753.62 ± 93.96/(22) | 1553.37 ± 48.56/(20) |
| Aldehydes | 1180.72 ± 25.82/(14) | 1384.15 ± 29.09/(13) |
| Furans | 65.52 ± 0.76/(1) | 97.49 ± 2.75/(2) |
| Ketones | 301.95 ± 7.74/(9) | 407.67 ± 9.03/(7) |
| Pyrazines | 806.85 ± 6.18/(6) | 1066.57 ± 11.79/(4) |
| Others | 410.30 ± 3.04/(7) | 459.35 ± 10.71/(8) |

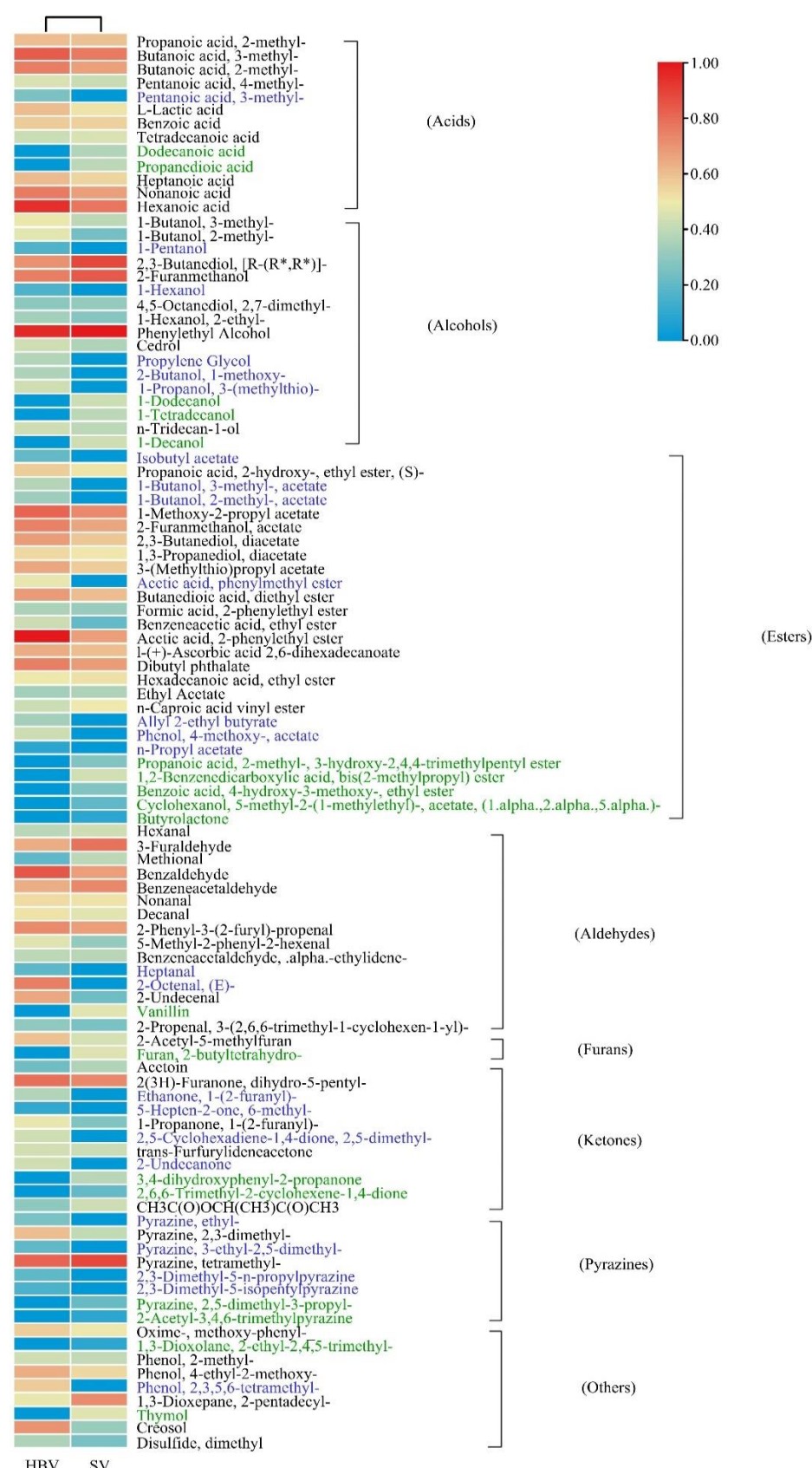

**Figure 4.** Heatmap of VFCs from aged HBV and SV. The black fonts indicate the common VFCs, while the blue and green fonts indicate the unique VFCs in both of aged HBV and SV, respectively.

The major VFCs in aged HBV and SV are shown in Table 5. Among them, 3-methylbutanoic acid, [R-(R*, R*)]-2,3-butanediol, phenylethyl alcohol, acetic acid 2-phenylethyl ester, benzaldehyde and tetramethylpyrazine, as main VFCs, were also found in SAV [21], possessing a cheesy [51,52], fruity [53,54], peachy [55], almondy [56], and nutty [21] smell, respectively. We also discovered that the contents of hexanoic acid, 2-furanmethanol acetate, acetic acid 2-phenylethyl ester, benzaldehyde, (E)-2-Octenal, 2-undecenal, and creosol were higher in HBV than those in SV (Table 4), while other VFCs were higher in SV than those in HBV. The SAV VFCs mainly came from the raw materials and the fermentation, heating, and aging process [57,58]. The VFC differences between BHV and SV were mainly produced by the components of the raw materials and their derivatives in the fermentation, heating and aging process since, in this study, the production process for HBV and SV was identical. In the future, the VFC differences in BHV and SV should be further studied.

**Table 5.** Summary of main VFCs in aged HBV and SV.

| VFCs | | Concentrations (mg·L$^{-1}$) | |
|---|---|---|---|
| | | HBV | SV |
| Acids | 3-Methyl- butanoic acid | 563.83 ± 10.40 | 458.56 ± 13.12 |
| | Hexanoic acid | 808.51 ± 6.02 | 431.15 ± 4.84 |
| Alcohols | [R-(R*,R*)]-2,3-Butanediol | 248.85 ± 3.43 | 1043.19 ± 30.54 |
| | 2-Furanmethanol | 411.11 ± 7.15 | 747.36 ± 10.15 |
| | Phenylethyl alcohol | 1578.52 ± 34.31 | 2478.52 ± 55.52 |
| Esters | 2-Furanmethanol acetate | 231.23 ± 14.30 | 105.97 ± 4.22 |
| | Acetic acid 2-phenylethyl ester | 1193.05 ± 32.56 | 495.92 ± 7.01 |
| Aldehydes | 3-Furaldehyde | 95.88 ± 2.74 | 367.65 ± 4.70 |
| | Benzaldehyde | 414.71 ± 3.33 | 303.41 ± 2.41 |
| | Benzeneacetaldehyde | 114.65 ± 3.29 | 203.71 ± 2.62 |
| | (E)-2-Octenal | 208.86 ± 5.56 | - |
| | 2-Undecenal | 84.14 ± 0.81 | 5.06 ± 0.75 |
| Ketones | Dihydro-5-pentyl-2(3H)-furanone | 174.52 ± 3.20 | 306.76 ± 4.78 |
| Pyrazines | Tetramethylpyrazine, | 722.33 ± 5.17 | 1038.16 ± 11.43 |
| Others | 2-Pentadecyl-1,3-dioxepane | 30.31 ± 0.49 | 221.92 ± 4.89 |
| | Creosol | 146.03 ± 4.26 | 50.82 ± 1.99 |

Note: "-" means not detected by HS-SPME-GC-MS.

## 4. Conclusions

In this study, HBV was made, using HB as the raw material, based on the process of SAV, a famous vinegar in China, in which sorghum is usually used as the raw material. The results showed that HB may be more appropriate than sorghum as the raw material, because the contents of starch and protein, the main components in brewing vinegar, were higher in HB. In addition, compared with SV, the acid and ester VFCs in HBV were superior.

**Supplementary Materials:** The following are available online at https://www.mdpi.com/article/10.3390/app11136039/s1, Table S1: VFCs in aged HBV and SV detected by HS-SPME/GC-MS.

**Author Contributions:** Conceptualization, F.C.; data curation, formal analysis and visualization, H.Z.; investigation, resources and software, J.Z. and F.L.; methodology and validation, Y.Z.; writing—review and editing and funding acquisition, F.C. All authors have read and agreed to the published version of the manuscript.

**Funding:** This research received no external funding.

**Institutional Review Board Statement:** Not applicable.

**Informed Consent Statement:** Not applicable.

**Data Availability Statement:** Not applicable.

**Conflicts of Interest:** The authors declare no conflict of interest.

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
