# Peer review of "Highland Barley Replaces Sorghum as Raw Material to Make Shanxi Aged Vinegar"

_applsci, doi:10.3390/app11136039_

Round 1

Reviewer 1 Report

The paper deals with an interesting topic: to replace one the main component for Shanxi Aged Vinegar, sorghum, by highland barley. Considering the importance of this type of vinegar in China, the possibility of substituting the main raw material by other cereal coming from areas with the need of improving their economy is specially interesting.

The paper is well written and supply all the information to make a proper evaluation of the proposed modification.

I only have some minor issues which should be addressed before the final acceptance.

1.- Could the authors check the information shown in lines 136-140.

2.- In line 218 I would not say “much higher” but only “higher”

3.- In line 249 authors say “…improved in…” making reference to Table 3. What it can be observed is an increase in concentrations, whether this an improvement or not it should be explained to keep the term “improved”.

4.- In Tables 4, 5 and S, data of concentrations instead of % should be given.

5.- Concentration of acetic acid should also be included.

Author Response

Response to Reviewer 1 Comments

The paper deals with an interesting topic: to replace one the main component for Shanxi Aged Vinegar, sorghum, by highland barley. Considering the importance of this type of vinegar in China, the possibility of substituting the main raw material by other cereal coming from areas with the need of improving their economy is specially interesting.

The paper is well written and supply all the information to make a proper evaluation of the proposed modification.

Response:Thank you very much for comments.

I only have some minor issues which should be addressed before the final acceptance.

1.- Could the authors check the information shown in lines 136-140.

Response: We have checked and modified the contents. Please see the blue words in Lines 137-141 of the revised manuscript (RMs).

2.- In line 218 I would not say “much higher” but only “higher”

Response:We have altered “much higher” into “higher” according to the reviewer's opinion.

3.- In line 249 authors say “…improved in…” making reference to Table 3. What it can be observed is an increase in concentrations, whether this an improvement or not it should be explained to keep the term “improved”.

Response:Thank you very much for your comments. We have added an explanation to make clearer. Please see the blue words in Lines ….. of RMs.

4.- In Tables 4, 5 and S, data of concentrations instead of % should be given.

Response:We have made correction according to the reviewer's opinion.

5.- Concentration of acetic acid should also be included.

Response:Since the detection results of acetic acid by HS-SPME/GC-MS have great inaccuracy due to the low efficiencies of solid phase extraction and desorption, acetic acid was not included. We have made an explanation in Lines… . of RMS.

Reviewer 2 Report

Dear authors, attached my few suggestions for improvement of the article:

Linea 35: the authors affirm that Highland barley (HB, Hordeum vulgare L.) is characterized by early maturity, cold resistance, stable yield and wide adaptability. Authors should indicate the amount of HB production during the last 5 years, useful to understand if it is enough for its use in vinegar.

Line 359 and line 360: remove indicationes to figure 2 and table 4 since figures and tables do not need to be mentioned in the conclusions.

Author Response

Response to Reviewer 2 Comments

The paper deals with an interesting topic: to replace one the main component for Shanxi Aged Vinegar, sorghum, by highland barley. Considering the importance of this type of vinegar in China, the possibility of substituting the main raw material by other cereal coming from areas with the need of improving their economy is specially interesting.

The paper is well written and supply all the information to make a proper evaluation of the proposed modification.

Response:Thank you very much for comments.

Linea 35: the authors affirm that Highland barley (HB, Hordeum vulgare L.) is characterized by early maturity, cold resistance, stable yield and wide adaptability. Authors should indicate the amount of HB production during the last 5 years, useful to understand if it is enough for its use in vinegar.

Response:Good question. We have added some information in Line…of RMS.

Line 359 and line 360: remove indicationes to figure 2 and table 4 since figures and tables do not need to be mentioned in the conclusions.

Response: We have deleted both of them.
